# Computation-Efficient Quantization Method for Deep Neural Networks

## Abstract

Deep Neural Networks, being memory and computation intensive, are a challenge to deploy in smaller devices. Numerous quantization techniques have been proposed to reduce the inference latency/memory consumption. However, these techniques impose a large overhead on the training procedure or need to change the training process. We present a non-intrusive quantization technique based on re-training the full precision model, followed by directly optimizing the corresponding binary model. The quantization training process takes no longer than the original training process. We also propose a new loss function to regularize the weights, resulting in reduced quantization error. Combining both help us achieve full precision accuracy on CIFAR dataset using binary quantization. We also achieve full precision accuracy on WikiText-2 using 2 bit quantization. Comparable results are also shown for ImageNet. We also present a 1.5 bits hybrid model exceeding the performance of TWN LSTM model for WikiText-2.

## 1 Introduction

Different variants of Deep Neural Networks have achieved state-of-the-art results in various domains from computer vision to language processing (Krizhevsky et al., 2012; Ren et al., 2015; Vaswani et al., 2017; Cho et al., 2014). However, newer models are becoming more memory and computation intensive to achieve performance improvements. For example, winner of ILSVRC 2015 ResNet (He et al., 2016) increased the number of layers by over 4x to gain less than 2% Top-1 accuracy improvement on ImageNet (Russakovsky et al., 2015). Compression techniques, such as knowledge distillation, pruning, low rank approximation and quantization, have been proposed to reduce the model size (Vapnik & Izmailov, 2015; Han et al., 2015; Sainath et al., 2013; Courbariaux et al., 2015). These compression techniques are evolving the field of model compression towards the goal of deploying DNN models on mobile-phone and other embedded devices.

Courbariaux et al. (2015) proposed the widely used technique for training quantized neural networks, where binary weights are used during forward and backward propagation, while full precision weights are preserved for accumulating gradients. Binary weights are approximated from full precision weights every iteration. (Zhou et al., 2018; 2017a) have proposed incremental quantization training procedure where the range for the weights is incrementally reduced. Choi et al. (2017) use Hessian weighted k-means clustering for quantization. Lee & Kim (2018) used iterative procedure of quantizing, de-quantizing and complete retraining of the full precision model, performed multiple times. All the techniques aimed to reduce the quantization error (error between full precision model and corresponding quantized model). However, most of these quantization techniques either adds extra set of hyper-parameters or modifies/lengthens the original training procedure.

Designing a neural network model consists of two main steps - choose a proper architecture/model given the characteristics of the task, and optimize the hyper-parameters for convergence and accuracy. Hyper-parameter search varies from number of layers in a model (Zoph et al., 2018) to learning rate (lr), batch-size (bs) combo (compare ResNet (He et al., 2016) and Inception (Szegedy et al., 2016) networks with altogether different hyper-parameter set). Courbariaux et al. (2015) requires updating the back-propagation procedure to train quantized networks. (Zhou et al., 2018; Lee & Kim, 2018) require very long training time because of multiple iterations of training and extra introduced hyper-parameters. Over time, focus on reducing the model size and the corresponding

inference latency has led to either lengthening or major modifications to training procedure. Our Proposed quantization technique addresses these issues resulting in easy adoption of our technique.

Our contributions include but are not limited to

- A simple quantization training method based on re-training without requiring major modifications to the original training procedure. Training consists of two phases: phase1 trains the full precision model (with quantization) and phase2 trains the binary model constructed by phase1.
- Reduce the overhead of expensive quantization techniques as quantization is performed only every few steps (specifically once every 500 iterations for the experiments).
- Maintained the total number of iterations and time required to train the quantized network compared to the full precision network.
- Achieve full precision accuracy for WikiText-2 and CIFAR dataset with 2-bit and 1-bit quantization respectively. Present a hybrid 1.5 bits LSTM models for WikiText-2 outperforming TWN LSTM model. Achieve performance comparable to existing works for ImageNet.

## 2  RELATED WORK

**Quantization.** Courbariaux et al. (2015) proposed the idea of training binary neural networks with quantized weights. Rastegari et al. (2016) introduced shared scaling factors to allow more range for binary values. (Hubara et al., 2016; Zhou et al., 2016; Lin et al., 2017; McDonnell, 2018; Hubara et al., 2018) built upon the training methodology along with introduction of binary activation units. Lee & Kim (2018) performs full precision retraining multiple times to train a quantized network. Ternary quantization was proposed (Zhu et al., 2017; Li et al., 2016; Wang et al., 2018) to mitigate the gap between full precision and quantized weight networks. Let $\mathbf{W} \in \mathbb{R}^{k \times c \times f \times f}$ represent a weight of a convolution layer $l$ with total n elements where $k, c, f$ represents output channels, input channels and size of the filter respectively. Rastegari et al. (2016) splits $\mathbf{W}$ into binary weight $\mathbf{B} \in \{-1, +1\}^{k \times c \times f \times f}$ and scaling factor $\boldsymbol{\alpha} \in \mathbb{R}^{+k}$ shared per output, where

$$\mathbf{B} = \text{sign}(\mathbf{W}) \qquad \boldsymbol{\alpha} = \langle \mathbf{B}, \mathbf{W} \rangle / n \tag{1}$$

obtained by minimizing $\|\mathbf{W} - \boldsymbol{\alpha}\mathbf{B}\|^2$. Binary quantization is extended to ternary where $\mathbf{B} \in \{-1, 0, +1\}^{n \times c \times k \times k}$. Ternary quantization introduces a threshold factor $\triangle_l$ to assign the ternary value to a weight. (Li et al., 2016; Zhu et al., 2017; Wang et al., 2018) have proposed various methodologies to evaluate the threshold. Lee & Kim (2018) performed ternary quantization by combining pruning and binary quantization.

Binary quantization was extended to multi-bit quantization using a greedy methodology by Guo et al. (2017). For k-bit quantization, minimizing $\|\hat{\mathbf{W}}_i - \boldsymbol{\alpha}_i \mathbf{B}_i\|$ for $i^{\text{th}}$ bit quantization resulted in,

$$\mathbf{B}_i = \text{sign}(\hat{\mathbf{W}}_i) \qquad \boldsymbol{\alpha}_i = \langle \mathbf{B}_i, \hat{\mathbf{W}}_i \rangle / n \qquad \text{where } \hat{\mathbf{W}}_i = \mathbf{W} - \sum_{j=1}^{i-1} \boldsymbol{\alpha}_j \mathbf{B}_j \tag{2}$$

referred as the greedy approach. Greedy approach was improved by refined method, where $\boldsymbol{\alpha}_i$ is computed by using $((\mathbf{B}_i^T \mathbf{B}_i)^{-1} \mathbf{B}_i^T \mathbf{W})^T$. Xu et al. (2018) improved refined method by performing a binary search on the given refined $\alpha$ set and alternately evaluating $\boldsymbol{\alpha}$ and $\mathbf{B}$. Low precision networks have also been proposed to reduce the gap with quantized activation units (Zhuang et al. (2018)). Quantization has also been applied to RNNs and Long Short Term Memory (LSTM) models as well (Hou et al. (2017); Guo et al. (2017); Zhou et al. (2017b); Xu et al. (2018); Lee & Kim (2018)). We use greedy quantization in this work due to its simple operations (although alternating yields the better results at the cost of higher computation overhead). Next section describes our quantization training procedure in detail.

## 3  ITERATIVE QUANTIZATION

Choromanska et al. (2015) shows that minima of high quality (measured by test accuracy) for large-size networks occur in a well-defined band. Choromanska et al. (2015) also conjectured that training

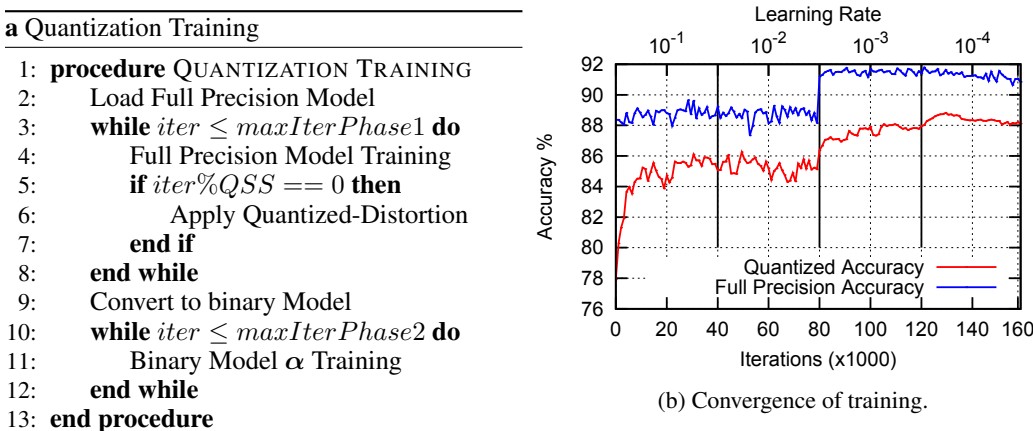

**a** Quantization Training

```
1:  procedure QUANTIZATION TRAINING
2:      Load Full Precision Model
3:      while iter ≤ maxIterPhase1 do
4:          Full Precision Model Training
5:          if iter%QSS == 0 then
6:              Apply Quantized-Distortion
7:          end if
8:      end while
9:      Convert to binary Model
10:     while iter ≤ maxIterPhase2 do
11:         Binary Model α Training
12:     end while
13: end procedure
```

(b) Convergence of training.

Figure 1: (a) Quantization Training algorithm (b) Convergence of accuracy using step training (Phase1) for ResNet32 on CIFAR-10 dataset.

using methods like stochastic gradient descent, simulated annealing converges to a minimum in the band. Minima in the band can have varying flatness, where flat minima have smaller error introduced to the accuracy upon adding distortion to the weight (Hochreiter & Schmidhuber (1995)). However, exploring through multiple minima has been a challenging task. Simulated annealing[1] (Kirkpatrick et al., 1983) explores through various minima, but does not aim to find wider minima. Motivated from simulated annealing, we propose a training technique to enable exploration of wider minimum among multiple minima. The technique allows escaping from relatively sharper minima and aims to find wider minima in the band. Our training procedure consists of two phases. Phase1 trains the full precision network with quantization. Phase2 fine-tunes the binary network obtained from phase1.

### 3.1 PHASE1: STEP TRAINING

The goal of phase1 is to produce a full precision model with optimized **B** and reduced quantization error (error between full precision and quantized model). Phase1 does not modify the original training procedure. Instead, in addition to the original training, phase1 just adds an extra distortion step using quantization (referred as quantized-distortion step), performed once every few iterations. Applying quantized-distortion to the weights consists of 3 parts - quantize the weights of each layer (quantization), convert the quantized weights back to full precision format (de-quantization), and update the full precision model with the de-quantized weights. Quantized-distortion is performed once every Quantized Step Size (QSS) iterations. Original training procedure combined with quantization-distortion is referred as Step Training (Figure 1a). Step training is performed in phase1 until the convergence of training (in principle).

Full precision training of the network for QSS iterations explores the curvature of the local convex surface of the current local minimum. Applying quantized-distortion post-training moves the model to the quantized lattice point on the training contour. Suppose that the quantized lattice point exist outside the curvature around a sharp minimum. Then, the network escapes such sharper minima in phase1. In contrast to existing quantization training methods, step training does not store quantized weights. Instead, step training updates and replaces full precision weights with their quantized version every few iterations.

**Quantization Step Size.** QSS determines the amount of retraining to be done between two quantized-distortion steps. QSS needs to be big enough to compensate for the error added by the distortion and let the network explore the current local curvature. However, QSS should not be too large to diverge the weights far away from a nearby quantized lattice point. Comparing big vs

---

[1]Simulated Annealing explores neighbors of the current solution in a randomized order. Worse neighboring solutions are also explored with relatively high probability (temperature) in the start (exploration). Temperature is reduced over time and later, only nearby better solutions are explored (exploitation).

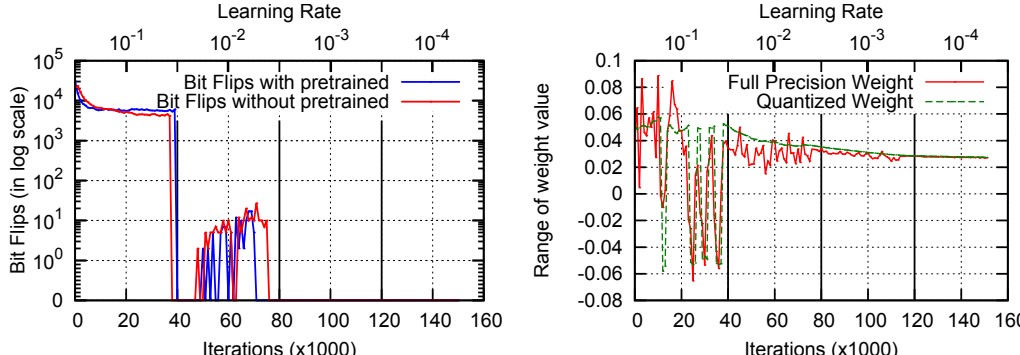

Figure 2: (left) Histogram of bit flips for all the weights of ResNet32 for CIFAR-10. (right) weights flipping their signs over the course of step training. Weight is randomly chosen from layer 30 of ResNet32 for CIFAR-10. Larger learning rate allows for more exploration and flipping of weights while small learning rate allows for fine-tuning and final convergence.

small QSS - big QSS allows the weights to explore farther allowing the binary representation of the weights to change (weights need large amount of updates to change their sign). On the other hand, small QSS allows the training to exploit the current local curvature and fine-tune $\boldsymbol{\alpha}$.

We observe that for a training procedure with fixed learning rate, starting with a big QSS and reducing QSS over the training period results in better convergence. However, the same behavior can be approximated with a fixed QSS and a varying learning rate. Figure 1b shows the movement of accuracy using step training with step-wise reducing learning rate and fixed QSS for ResNet32 on CIFAR-10. Larger learning rate enables larger amount of updates (and hence more curvature exploration) given the same gradient from the back propagation (fluctuations in the accuracy). On the other hand, smaller learning rate helps exploit (fine-tune the parameters inside the current minimum) as shown by a smoother rise in accuracy. Hence, we use fixed QSS (500 iterations) in the rest of the manuscript, although one can use varying QSS for further fine-tuning.

**Convergence of B.** We observe that **B** converges earlier compared to $\boldsymbol{\alpha}$ during step training. Such an observation is demonstrated in our experiment with step training for CIFAR-10 with ResNet32. Figure 2 shows the movement of weight with step training[2]. Initially, the sign bits of the weight flip frequently (with higher learning rate). However, with smaller learning rate (after 80K iterations for CIFAR-10), **B** do not change and only $\boldsymbol{\alpha}$ is optimized.

**Convergence of $\boldsymbol{\alpha}$.** Let **W** be a tensor of full precision weights with n elements. **W** is quantized into **B** (binary tensor) and $\boldsymbol{\alpha}$ (shared scaling factors). Step training updates **W** every iteration. On the other hand, $(\mathbf{B}, \boldsymbol{\alpha})$ are calculated every QSS iterations. Let $\triangle\mathbf{W}$ denote the total update accumulated for **W** since the last distortion step. Let $\triangle\boldsymbol{\alpha}$ denote the change between the new $\boldsymbol{\alpha}$ and the $\boldsymbol{\alpha}$ calculated at previous distortion step. For binary quantization, the updated $\boldsymbol{\alpha}$ is given by-

$$\boldsymbol{\alpha} + \triangle\boldsymbol{\alpha} = \frac{1}{n} \sum |\mathbf{W} + \triangle\mathbf{W}| \tag{3}$$

where $|.|$ is the absolute function. Starting from a common absolute quantized value ($\boldsymbol{\alpha}$), the weights sharing the same $\alpha$ update independently ($\forall i, j \, w_i, w_j \in \mathbf{W}, \partial w_i/\partial w_j = 0$). With large QSS, as the weights diverge from $\alpha$, the update for $\alpha$ becomes inefficient and noisy. Although phase1 results in optimized **B**, phase1 does not completely optimize $\boldsymbol{\alpha}$ (within the limited number of iterations). Need for improved convergence of $\boldsymbol{\alpha}$ forms the motivation for phase2.

## 3.2  PHASE2: $\boldsymbol{\alpha}$ TRAINING

Phase2 starts by converting the full precision trained model from phase1 to the corresponding binary model. The full precision weights **W** in phase1 are replaced with the corresponding binary version, $\boldsymbol{\alpha}$ and **B** in the model. **B** is fixed and only $\boldsymbol{\alpha}$ is trained. Phase2 only constructs the binary model

---

[2]All the figures are presented with 160k iterations, with learning rate decayed by 0.1 every 40k steps to investigate the effects of 4 different learning rates.

and does not construct the full precision model. Phase2 is faster compared to phase1 due to fewer training parameters, use of binary weights, and no quantized-distortion step. Phase2 is performed with a smaller learning rate after the bit-flips do not occur anymore in phase1. Similar to phase1, phase2 also uses the original training procedure but with fewer number of trainable parameters.

In the complete training procedure, we first perform phase1 followed by phase2. The trained binary model at the end of phase2 represents the output of the complete training procedure. This complete training procedure uses the same number of iterations as that in the original training procedure. As a result, the total training time combining phase1 and phase2 is equivalent to the original full precision training time.

### 3.3 Special Care for CNNs

This section compares different ways to apply quantization on a 4D tensor kernel. Quantization for a 2D weight matrix $W$ with n elements outputs a binary 2D matrix $B \in \{-1, +1\}^n$ and shared scaling factor $\alpha$ per row of $W$. All the weights in a row share the same $\alpha$, where $\alpha \in \boldsymbol{\alpha}$. Further, a row in the matrix can be split into $t$ sub-rows (referred as tables), where each table has a different $\alpha$. For quantizing a 4D tensor kernel $\mathsf{W} \in \mathbb{R}^{k \times c \times f \times f}$ in a convolution layer, $\mathsf{W}$ is reshaped to 2D matrix $W \in \mathbb{R}^{k \times cff}$ (Rastegari et al., 2016) ($k$ is the number of output features, $c$ is the number of inputs features and $f$ is the filter size). There are total $k$ $\alpha$, each shared by $c \times f \times f$ number of weights. Each output feature in the convolution layer has $c \times f \times f$ weights. Thus, there is only 1 $\alpha$ shared by all the weights for an output feature. As the quantized weights can only take the value of $\{+\alpha, -\alpha\}$, all the inputs for an output feature can only be weighted by the same absolute factor $\alpha$. Hence, the representative power of the quantized network is limited.

To alleviate this problem, we convert the 4D tensor $\mathsf{W}$ to 2D matrix differently. $\mathsf{W}$ is transposed to $f \times k \times c \times f$ and then reshaped to 2D matrix $\mathbb{R}^{f \times kcf}$ (referred as skewed matrix). Next, each row of size $k \times c \times f$ is split into $k/f$ sub-rows. Each sub-row has a different $\alpha$. Total number of $\alpha$ remains the same as above ($k$). Furthermore, the inputs for an output feature can now be weighted by $f$ number of unique $\alpha$. Note that some $\alpha$ will be shared among different output features as well. In our experiments, skewed matrix shows better results. Section 5.1 shows the benefit in accuracy using the skewed matrix for quantization.

## 4 K-means Loss and Shuffling

This section aims to limit the divergence of weights from each other during training to get more accurate estimation of $\alpha$. $L_2$ regularization (in the form of $L_2$ loss) is frequently used in training of neural networks. $L_2$ loss prevents the weights from exploding and suppresses the magnitude of the weights. However, $L_2$ loss does apply any restriction on the variance of the weights. As $\boldsymbol{\alpha}$ is obtained using Equation 1, higher variance in weights results in higher quantization error. We aim to reduce quantization error by introducing $L_{KM}$ loss function to reduce the variance of the weights. Let $\mathsf{W}_i$ represent all the weights in a layer $i$. Let $\boldsymbol{w} \in \mathsf{W}_i$ be a subset of weights sharing common $\alpha$ ($\boldsymbol{w}$ are referred as clusters from now). The new loss is represented as:

$$L_{KM} = \frac{c}{\|\mathsf{W}\|^0} \sum_{\forall \boldsymbol{w} \in \mathsf{W}} \|\boldsymbol{w} - \mathrm{avg}(|\boldsymbol{w}|)\|^2 \tag{4}$$

where c is a constant, avg(.) computes the average of the inputs. $L_{KM}$ divides the weights $\mathsf{W}$ into different clusters $\boldsymbol{w}$ with common $\alpha$ and limits the divergence of weights from the cluster average of its absolute values $\boldsymbol{\alpha}$. $L_{KM}$ restricts the independent movement of weights. Similar with $L_2$ loss, a diverged weight increases the $L_{KM}$. In addition, a diverged weight also shifts the cluster average, increasing the $L_{KM}$ loss further. Thus $L_{KM}$ encourages a lower variance in the weights with common $\alpha$ and improve quantization as a result. The constant factor $c$ is set to be the same as weight decay rate (the constant for $L_2$ loss is reduced to mitigate the effect of $L_2$).

**Shuffling.** $L_{KM}$ can be extended for multiple tables, where a row of a quantized matrix has multiple shared $\alpha$. With multiple tables, let $\boldsymbol{w}$ correspond to a subset of row, where the subset shares the common $\alpha$. $L_{KM}$ helps in better approximation for $\boldsymbol{\alpha}$ by forcing a predetermined group of weights to exhibit low variance. We could also achieve a better approximation for $\boldsymbol{\alpha}$ by re-arranging the

weights so that similar values are grouped together. Note that rearranging is applicable for DNNs with multiple tables only.

Let the weight matrix between two fully connected layers $l^i$ and $l^{i-1}$ be represented by $\boldsymbol{W}^{i,i-1}$. Two nodes in a layer $l^i$ given as $l^i_j$ and $l^i_k$ can be swapped by switching the rows $j,k$ of weight matrix $\boldsymbol{W}^{i,i-1}$ and the columns $j,k$ of $\boldsymbol{W}^{i+1,i}$. The layout of the weight matrix can be set to cluster the desired weights without impacting the output of the network. Swapping the nodes in layer $l^i$, $l^{i-1}$ swaps the rows and columns of $\boldsymbol{W}^{i,i-1}$ respectively. Thus, nodes in each layer can be swapped independently. K-means clustering is used to find an optimized configuration of weights in terms of grouping similar values together first. And then the nodes in the layer are swapped to enforce such configuration, reducing the quantization error. The methodology of finding and applying the optimal swapping configuration is termed as shuffling of nodes.

Because applying shuffling of nodes to the current layer requires a layer before and after the current layer, we introduce a shuffle layer in the start and end of the network to allow shuffling in first and last layer of the network. The shuffle layer stores the shuffle configuration and behaves as a mapping layer. The overhead of the shuffle layer is less than 1% of the model size (same as the size of bias in a layer). Shuffling is applied in the weight distortion step in phase1, when quantizing the weights.

## 5 EXPERIMENTS

Experiments are performed with CNNs and RNNs to show the effectiveness of the proposed method. All the experiments are performed using Tensorflow (Abadi et al., 2016) using 2 Titan X GPUs. Full precision models for CNNs are obtained from tensorflow models repository[3], while RNN models are obtained from Verwimp et al. (2017)[4]. We quantize all the layers of the network unless specified otherwise. Iterative quantization training is performed on pre-trained full precision models. Greedy quantization using Equation 1 is used for all the experiments. Quantization Step Size is set to 500 constant throughout all the experiments.

### 5.1 CIFAR

ResNet32 is trained on CIFAR-10 (Krizhevsky, 2009) for 90k iterations, where the first 60k iterations are performed using step training and remaining 30k iterations are performed with $\boldsymbol{\alpha}$ training. 60% pruning rate is set for ternary quantization. Training ResNet32 using our proposed quantization training method does not incur any increase in training time.

Table 1 shows the improvement by combining the techniques discussed in previous sections in an incremental manner for ResNet32. We use the $k$ $\alpha$ for quantization following Rastegari et al. (2016) as default quantization mode ($k$ is the number of output features for a convolution layer). Different QSS schedules were tried where QSS starts with a high value and is reduced over the training procedure. QSS schedule produces better accuracy compared to fixed QSS for step training. Note that, however, $\boldsymbol{\alpha}$ training eliminates the need to fine-tune over the QSS and achieves the same accuracy. Number of tables for skewed mode have been set to have the same model size as default mode (resulting in the same number of $\boldsymbol{\alpha}$).

Table 2 provides our final accuracy for CIFAR dataset with all the techniques combined using ResNet32 and WideResNet 28-10 models (70x bigger model size compared to ResNet32). Our model provides similar performance compared to TTQ (Zhu et al., 2017) using ResNet32. Our method achieves full precision accuracy for WideResNet 28-10 on both CIFAR-10 and CIFAR-100, compared to the results by McDonnell (2018) without changing the training procedure. We believe that WideResNet demonstrates smaller quantization error than ResNet32 because Li et al. (2018) reported that wider networks facilitates flatter minima.

### 5.2 WIKITEXT-2

LSTM model with 1 layer consisting of 512 nodes is used for WikiText-2 (Merity et al., 2017) dataset, with same hyper-parameter settings as followed by Xu et al. (2018). Performance is mea-

---

[3]https://github.com/tensorflow/models/
[4]https://github.com/lverwimp/tf-lm

Table 1: Accuracy improvement by each method incrementally for ResNet32 on CIFAR-10.

| Config | Accuracy % |
|---|---|
| Full Precision | 92.47 |
| Binary Step Training (BST) | 88.18 |
| BST without pre-trained | 88.09 |
| Ternary Step Training (TST) | 89.27 |
| TST + QSS schedule | 90.45 |
| TST + $\alpha$ training | 90.4 |
| TST + skewed matrix | 91.3 |
| TST + skewed matrix + $L_{KM}$ | 91.8 |
| TST + skewed matrix + $L_{KM}$ + $\alpha$ training | 92.36 |

Table 2: Accuracy Comparison for CIFAR using ResNet32 and WideResNet for TWN and binary models. TTQ: Zhu et al. (2017) (TWN model), Wide-1b: McDonnell (2018) (binary model)

| Config | Accuracy % | | | | | |
|---|---|---|---|---|---|---|
| | CIFAR-10 ResNet32 | | CIFAR-10 WideResNet | | CIFAR-100 WideResNet | |
| | Ours | TTQ | Ours | Wide-1b | Ours | Wide-1b |
| Full Precision | 92.47 | 92.33 | 95 | 95.77 | 78.3 | 81.37 |
| Binary Model | 92.36 | 92.37 | 95.02 | 95.54 | 78.3 | 81.06 |

sured with Perplexity Per Word metric (PPW). Full precision PPW is 100.2. Activation quantization requires quantization to be performed every iteration for inference, wiping out the speed up obtained with quantized weights for inference. Activation quantization slows down training as well. Thus, we use 32bit activations while 3bit activations used by Xu et al. (2018). Our 2-bit alternating quantization (greedy quantization replaced with alternating quantization in our proposed method) reaches full precision PPW (Table 3).

Table 3 compares the accuracy for multi-table models (multiple $\alpha$ per row of the matrix to be quantized) with TWN model (TWN method from Li et al. (2016) combined with our training method). Our multi-table model (8 tables for Embedding and Softmax layer, 16 tables for LSTM layer) combined with $L_{KM}$ loss function generates PPW equivalent to TWN PPW. Multi-table model accounts to 1.5 bits per weight in total (after accumulating all the $\alpha$ and $B$). Applying $L_{KM}$ reduces the model size by 25% from TWN (2 bit) to 1.5 bits with equivalent PPW. We also perform 1 bit quantization reaching 128.18 PPW.

**Hybrid LSTM.** In the WikiText-2 LSTM model, Embedding and Softmax layer each forms over 45% of the full precision model size. Therefore, we selectively optimize the number of quantization bits for each layer to achieve higher compression rate. 1 bit quantization was found to be sufficient for Embedding layer. However, other layers required more number of bits. We fix 1bit quantization for Embedding layer (1 table per row), TWN for Softmax layer and vary the number of quantization bits for LSTM layer. Using 2bit for LSTM layer (1.53 bits per weight in total) provides PPW better than our TWN greedy model, with 25% smaller model size.

## 5.3 ABLATION STUDY

**Random Initialization.** The distribution of bit-flips and convergence of accuracy for step training with randomly initialized model and with pre-trained model is observed to be similar (Figure 2). The accuracy gap between Binary Step Training model with pre-trained model and BST without pre-trained model less than 0.1% (Table 1).

**One-Step Quantization.** We experiment the degradation in accuracy with just one-step quantization (no retraining). ResNet32 with full precision accuracy of 92.47% on CIFAR-10 produces 44.33% accuracy. To examine the potential of $L_{KM}$, ResNet32 is again trained with random initialization in full precision mode with $L_{KM}$ (without any form of quantization). Although, the full

Table 3: Comparison of Perplexity Per Word (PPW) for LSTM models on WikiText-2 dataset. Multi-table models use 1 quantization bit with multiple tables (8 tables (8 $\alpha$ per row) for Embedding and Softmax layer, 16 tables for LSTM layer). Hybrid models use 1 bit quantization for Embedding layer and TWN quantization for Softmax layer. 2 to 4 quantization bits for LSTM layer provides 1.53 to 1.65 bits per weight models. Results for Guo et al. (2017) are taken from Xu et al. (2018).

| 2-bit models | PPW | 1.5-bits models | PPW | Hybrid models | PPW |
|---|---|---|---|---|---|
| Guo et al. (2017) | 105.8 | Multi-table | 117.13 | 1.53 bit | 108.1 |
| Xu et al. (2018) | 102.7 | Multi-table + $L_{KM}$ | 115.26 | 1.6 bit | 105.46 |
| Our Greedy | 104.15 | Multi-table + Shuffle | 116.08 | 1.65 bit | 103.58 |
| Our Alternating | 100.3 | TWN | 115.05 | | |

Table 4: Robustness of Step Training to Quantization Step Size for CIFAR-10 with ResNet32. Accuracy varies within a range of 3% with QSS ranging from 10 to 2500.

| Quantization Step Size | 10 | 50 | 100 | 500 | 1000 | 2500 | 5000 | 10000 |
|---|---|---|---|---|---|---|---|---|
| Accuracy % | 86.77 | 88.03 | 88.97 | 89.15 | 88.88 | 86.52 | 83.01 | 78.54 |

precision accuracy drops to 91.8%, one-step quantization accuracy goes up to 76.32%. Increasing the regularization constant for $L_{KM}$ yields the one-step quantization accuracy as 84.51%.

**Robustness.** Table 4 shows the robustness of iterative quantization with varying QSS over 2x in the order of magnitudes. As explained in section 3.1, varying learning rate can provide the same functionality as varying QSS. As most of the modern neural networks use special learning rate policy (such as exponential decay, step-wise decay), the training procedure is overall robust to the choice of QSS. The simplicity of the algorithm and robustness to the added hyper-parameter facilitate quick adoption of our proposed technique.

## 5.4 IMAGENET

Full precision ResNet18 is trained on ImageNet (Russakovsky et al., 2015) following the base repository[3] with batch size of 256. Our binary model reaches 60.6% Top1 accuracy compared to full precision accuracy of 69.6% (Table 5). Our model shows comparable accuracy compared to existing quantization methods. We believe our model can reach higher accuracy by using layer-by-layer quantization as done by Zhou et al. (2018).

## 5.5 QUANTIZATION OVERHEAD

Let training time per iteration be defined as the combined time to perform forward-propagation and back-propagation on a batch of data. We evaluate the overhead of performing a quantization step once relative to the defined training time per iteration. All the timings are averaged over 1000 iterations, averaged over ResNet32 and WideResNet. We observed that overhead of using greedy quantization is the lowest (8% and 12% of the training time for 1bit and 2bit quantization). More sophisticated quantization methods using regression or iterative procedures, namely refined and alternating quantization, have overhead of 5x and 40x respectively over the training time. Table 3

Table 5: Accuracy Comparison for Imagenet using ResNet18 for 1 bit quantization.

| Config | Top1 Accuracy |
|---|---|
| Full Precision | 69.6 |
| Li et al. (2016) | 57.5 |
| Dong et al. (2017) | 58.36 |
| Our binary model | **60.6** |
| Rastegari et al. (2016) | 60.8 |
| Zhou et al. (2018) | 64.72 |

compares the benefit of using these quantization methods, where alternating quantization shows the best performance despite the biggest overhead. As our training method, unlike existing methods, performs quantization once every 500 iterations, the overhead of the quantization is reduced by 500x. As a result, the overhead of the most expensive quantization remains to be 10% of the training time.

## 6 CONCLUSION

In this work, we have presented an iterative quantization technique performing quantization once every few steps, combined with binary model $\alpha$ training. Step training explores flatter minima while escaping sharp minima and $\alpha$ training performs exploitation of the chosen minima. We also presented a loss function $L_{KM}$ which allows weights to be adjusted for improved quantization. We demonstrated full precision accuracy recovery with CIFAR and WikiText-2 dataset with our quantized models. We also presented a hybrid model with 1.5 bits performing better than the our TWN model.

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

# APPENDIX

## A UPDATE TO $\alpha$ IN PHASE2

For lower learning rates, as **B** are fixed, $(w \cdot (w + \triangle w)) > 0)$, where $w \in \textbf{W}$. Equation 3 is correspondingly updated to -

$$\alpha + \triangle\alpha = \frac{1}{n}\sum |\textbf{W} + \triangle\textbf{W}| = \frac{1}{n}\sum |\textbf{W}| + \frac{1}{n}\sum \triangle\textbf{W} \circ \textbf{B} \qquad (5)$$

whewre $\circ$ is Hadamard product. Equation 5 shows that updates to **W** are directly merged into $\alpha$ (after **B** has converged). Thus, the training procedure for quantization with low learning rate can be more efficient by optimizing $\alpha$ only, compared with optimizing both **B** and $\alpha$. Phase2 presents such an efficient optimization method.

## B TRAINING DETAILS

We provide more details on the training procedure for the networks for all the datasets.

### B.1 CIFAR

CIFAR dataset consists of 50000 training images with 10000 test images. Each image is of size 32x32. CIFAR-10 dataset classifies the corpus of images into 10 disjoint classes. CIFAR-100 classifies the image set into 100 fine-grained disjoint classes.

**ResNet.** ResNet32 and WideResNet 28-10 were both trained for 90k iterations with batch size of 128. Step-wise decay learning schedule was used. With initial learning rate of 0.1, learning rate was decayed with 0.1 at 40k, 60k and 80k iterations each. Momentum training optimizer was used for training with momentum set 0.9. 0.0005 was set as weight decay rate. Training was pre-processed with random cropping and random horizontal flipping. Evaluation data was pre-processed with a single central crop only. Quantization Step Size was set as 500 during step training.

**Pruning.** ResNet32 was pruned with 60% as the final sparsity, with an initial sparsity of 20%. Pruning was started with a pre-trained model. Pruning was gradually increased at en exponential rate (exponential factor of 3) with the pruning being performed every 100 iterations. Re-training for pruning for performed for 40k iterations.

### B.2 WIKITEXT-2

WikiText-2 contains 2088k training, 217k validation and 245k test tokens, with a vocabulary of 33k words. The model for Wikitext-2 consisted of 1 LSTM layer with 512 units. Initial learning rate was set as 20. Learning rate was decayed by 1.2 every 2 epochs. Training was terminated once the learning rate was less than 0.001 or maximum of 80 epochs was reached. The absolute gradient norm was set at 0.25. The network was unrolled for 30 time steps. Training was performed with a dropout ratio of 0.5. Weights were clipped to an absolute maximum of 1.0. Quantization Step Size was set as 500 during step training.

**Divergence with Greedy Quantization.** Using greedy quantization with 2bit quantization for LSTM model always diverged the training with WikiText-2 dataset. To make the model converge up to some extent, we used 1bit quantized model as an initialization for 2 bit quantization. Although, 1bit initialized model converges for a few epochs but also diverges after 10-15 epochs. The results reported in Table 3 for 2 bit greedy quantization follow initializing with 1bit quantized model. The divergence in network with greedy quantization is the reason for using TWN for Softmax layer (and not 2bit quantization) in our hybrid model.

### B.3 IMAGENET

ImageNet consists of 1281176 training images with 50000 validation images, classified into 1000 classes. ResNet18 network was used for training. Training data was pre-processed with random

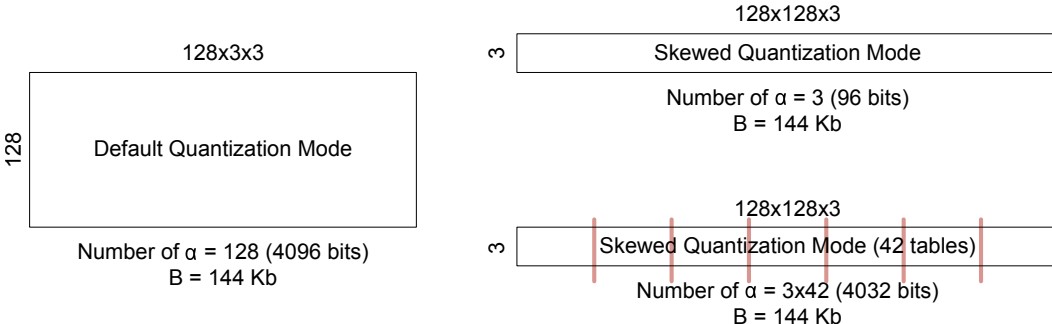

Figure 3: Comparing different skewed and default quantization mode (Rastegari et al., 2016) with multiple tables. Skewed quantization mode converts a 4d tensor $\mathbf{W} \in \mathbb{R}^{k \times c \times f \times f}$ into $\mathbb{R}^{f \times kcf}$, and default quantization mode convert into $\mathbb{R}^{k \times cff}$ (where k is number of output features, c is number of input features, fxf is the filter size). For a kernel of shape 128x128x3x3, skewed mode with 42 tables has a memory footprint equivalent to default quantization mode.

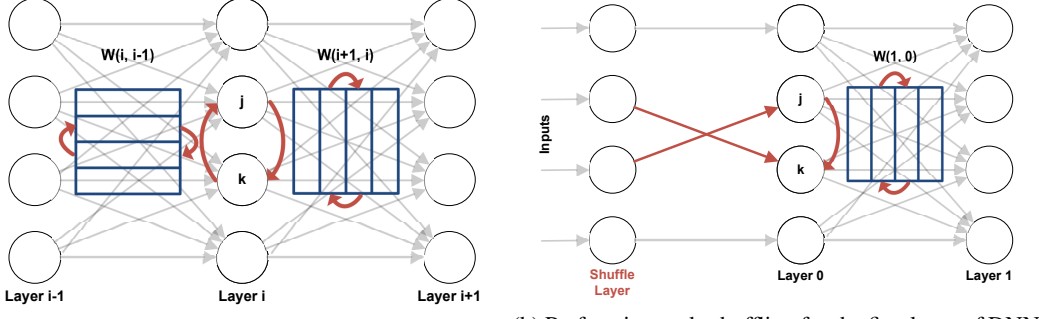

(a) Performing node shuffling for a layer in DNN    (b) Performing node shuffling for the first layer of DNN

Figure 4: Performing node shuffling for a layer in DNN. (a) Node shuffling for a layer in between 2 layers is performed by switching the row and columns of the previous and next weight matrix. (b) Node shuffling for the first layer of DNN is performed using shuffle layer. Shuffle layer maps the input to match the shuffling order with a very small overhead. Shuffle layers can also be added for RNN/LSTM layer to performing shuffling of layers independently in the weight matrices of RNN/LSTM layer.

cropping and random horizontal flipping. However, validation data was pre-processed with 1 single central crop. Step-wise decay learning rate schedule was followed with initial learning rate of 0.1 and decayed at epochs 30, 60, 80 and 90 by a factor of 0.1. The complete training procedure was performed for 100 epochs with a batch size of 256. Momentum training optimizer was used for training with momentum set 0.9. 0.0005 was set as weight decay rate. Quantization Step Size was set as 500 during step training.

## B.4    BATCH NORMALIZATION

Batch normalization (Ioffe & Szegedy, 2015) parameters ($\mu$ and $\sigma$) are updated using moving average. Consequently, the effect of quantized-distortion performed even 100 iterations earlier would have less than 1% effect on the BN parameters (with momentum=0.99). As a result, BN parameters are not suited well for quantized model and results in drop in evaluation accuracy for the quantized model. To avoid the drop in evaluation accuracy by BN, the BN parameters are re-evaluated over 1 train epoch (keeping the other parameters fixed) before performing evaluation for the phase1. Phase2 does not require any special care for batch normalization as there is no distortion step.

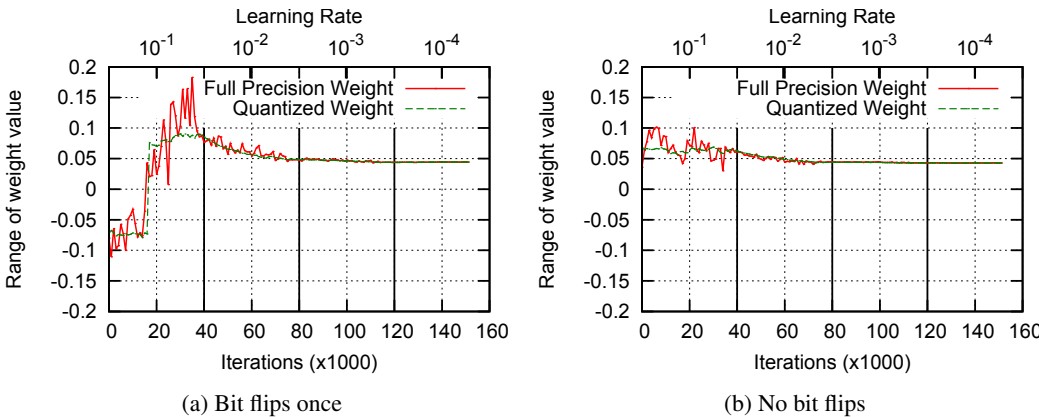

(a) Bit flips once

(b) No bit flips

Figure 5: Convergence of weight when training using Step Training. Compares two scenarios where (a) bit flips once and (b) bit does not flips during the course of step training. Step training was performed for ResNet32 using CIFAR-10 dataset.

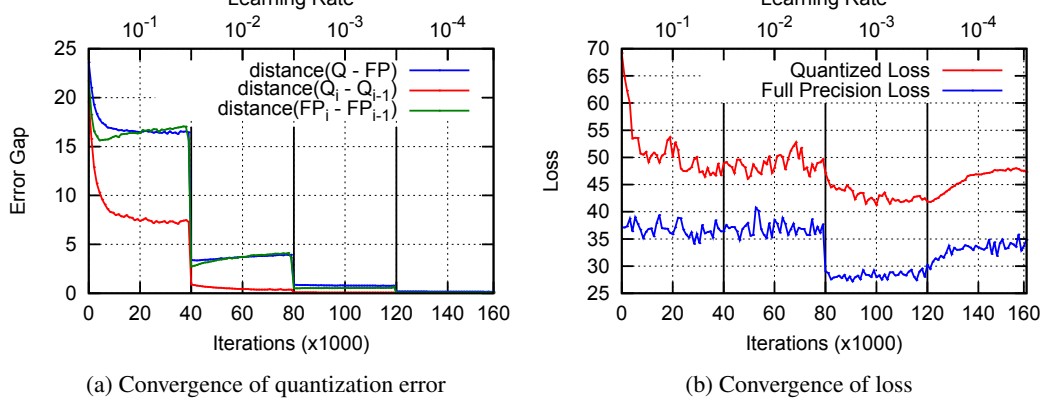

(a) Convergence of quantization error

(b) Convergence of loss

Figure 6: (a) Shows the convergence of quantization error with decrease in learning rate. The distance between consecutive quantization weights (QSS iterations apart) and distance between corresponding full precision weights is also shown. Distance between consecutive quantization weights is directly correlated with the number of weight flips (Figure 2). (b) Shows the movement of loss with step training for ResNet32 using CIFAR-10 dataset. The loss rises for the last learning rate as distortion causes more damage compared what training with the small learning rate can repair.

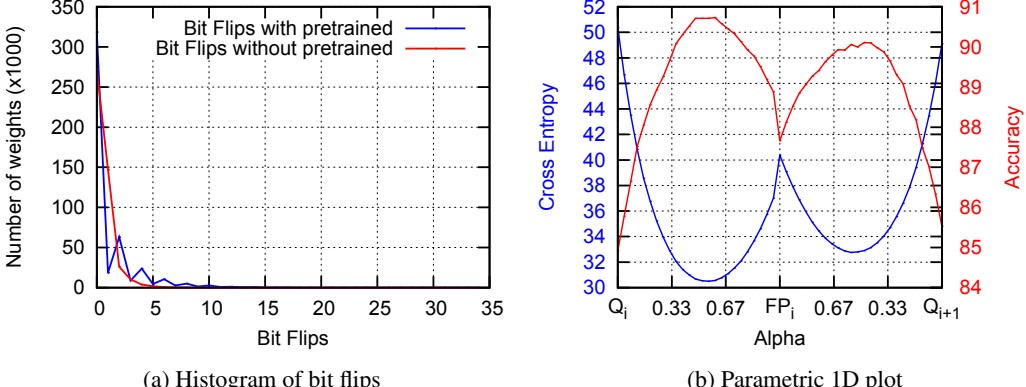

(a) Histogram of bit flips

(b) Parametric 1D plot

Figure 7: (a) Shows the same behavior of histogram of total bit flips for step training with or without pre-trained model for ResNet32 with CIFAR-10 dataset. (b) Parametric 1-D plots as described in (Goodfellow et al., 2015; Keskar et al., 2017). $Q_i$ denote the weight set after performing quantized-distortion for the i[th] time while performing step training. $FP_i$ correspond to the full precision weight set just before performing the i[th] quantized-distortion. The plot is for cross entropy along a line segment containing the two points. Specifically for $a \in [0, 1]$, we plot $f(aQ_i + (1 - a)FP_i)$. The same is plotted for $FP_i$ and $Q_{i+1}$.

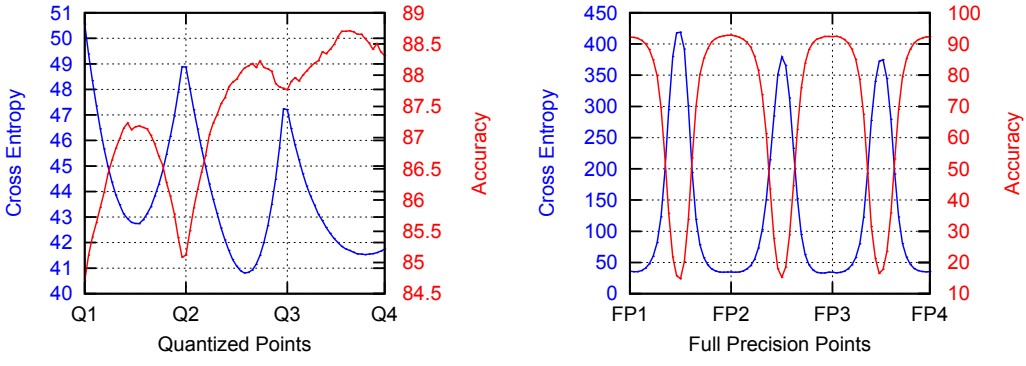

(a) Parametric 1D plot for quantized weights.

(b) Parametric 1D plot for full precision weights.

Figure 8: (a) Parametric 1-D plots quantized weight sets at different learning rates while performing step training. Step training is performed with 160k iterations, with learning rate decayed by 0.1 every 40k steps to investigate the effects of 4 different learning rates. All the quantized points are sampled randomly at different iterations at different learning rate. $Q_1$ is obtained at learning rate 0.1, $Q_2$ at 0.01, $Q_3$ at 0.001 and $Q_4$ at 0.0001. (b) Starting from these quantized weight sets, the model is retrained using full precision training method for the remainder of the iterations. This retraining gives us $FP_1, FP_2, FP_3, FP_4$. The rise in the loss function along the path between two full precision points shows the existence of full precision weights in different local minimum.

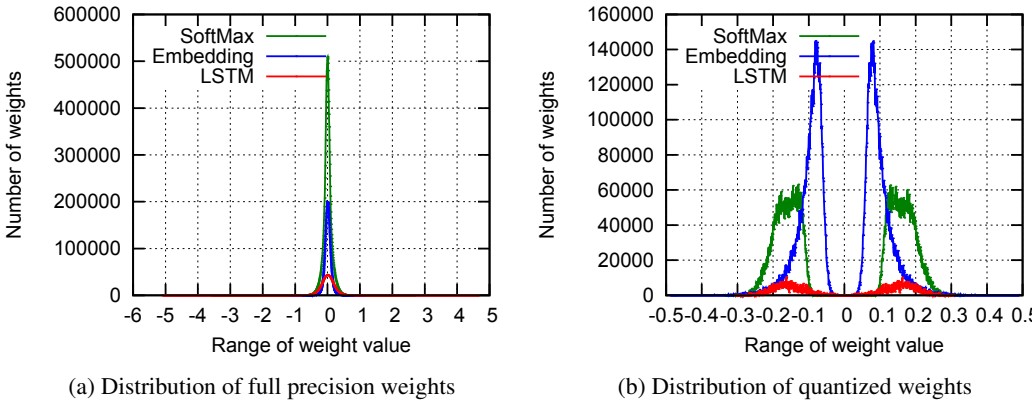

(a) Distribution of full precision weights

(b) Distribution of quantized weights

Figure 9: Distribution of weights of the WikiText-2 LSTM model for full precision and quantized trained model.

