# OpenReview forum: "Computation-Efficient Quantization Method for Deep Neural Networks"
_ICLR.cc/2019/Conference_

### Official Review · AnonReviewer3 · 2018-10-29
**An effective training methods for network quantization  was proposed, however, the claimed contribution has been well justified.**

**Rating:** 5
**Confidence:** 4

**Review:**

This paper proposes a method based on re-training the full-precision model and then optimizing the corresponding binary model. It consists of two phases: (1)  the full-precision model training where the quantization step is introduced through QSS to train the network, and (2) fine tuning of quantized networks, where  the trained network was converted into a binary model. In addition, using the skewed matrix for quantization improves the accuracy. Then a loss function based on the k means form is used to normalize the weight for reducing the quantization error. Quantization experiments for CNNs or LSTMs have been conducted on CIFAR10, CIFAR100, IMAGENET, and WikiText-2 dataset.

This paper has been presented clearly. However, it can be improved by introducing the motivation of the tricks(e.g. skewed matrix and loss related to k-means ) used for quantization.

In the experiments, the precision improvement on the CIFAR and ImageNet dataset performs worse than some competitors. For example, the precision of the proposed method was significantly worse than Zhou et al, 2018 on ImageNet. It is better to  analyze the reason.

In addition, as claimed from the introduction, the contribution of this paper was  to reduce the overhead of expensive quantization. However, no experimental results on computation time  and parameter size have been shown.

---

> ### Author Response · Authors · 2018-11-15
> **reply**
>
> Thank you for the review.
>
> We have updated the manuscript as per your suggestion to provide an introduction to the motivation for the section 3.3 and section 4.
>
> On cifar10 and cifar100 experiments with wide-resnet, our method recovers full accuracy with its baseline (baseline is taken from Tensorflow models implementation - https://github.com/tensorflow/models) as given in Table 2. With resnet32, our accuracy loss is 0.11%, with final accuracy matching that from TTQ. However, unlike TTQ, our occasional distortion reduces amount of computation overhead from quantization and does not require any intrusive change in the training procedure. Also refer to our response to AnonReviewer2 for more details on this.
>
> The greedy quantization has low overhead to the original training time (10% and 15% of the training time for 1bit and 2bit quantization). Using more sophisticated techniques like refined, alternating quantization leads more overhead (over 5x, 40x compared to the training time respectively). As table 3 shows the benefit of these quantization methods, our training method, unlike existing methods, performs quantization once every 500 iterations, nullifying the overheads of these quantization to less than 10% of the original training time. We have updated our manuscript accordingly as well.
> 1 bit quantization leads to almost 32x reduction in parameter size, while ternary and 2 bit quantization reduce parameters by almost 16x.
>
> Please refer to our response for AnonReviewer1 for our imagenet results in comparison with Zhou et al. (2018).

---

> > ### Comment · AnonReviewer3 · 2018-12-11
> > **Thanks for the response.**
> >
> > After reading the responses and the revised manuscript, the reviewer still did not find complexity analysis in Sec 5.5. The complexity analysis is important since the claimed contribution is to improve training quantitative models. There are still no comparison with other quantization methods in terms of computational time and memory.
> >
> > In addition, the explanation to the comparison to Zhou et al. is not convincing.  There is a significant gap between the results.
> >
> > The reviewer will keep the rating.

---

### Official Review · AnonReviewer1 · 2018-11-02
**This paper considers the problem of training weight quantized deep neural networks. An iterative method is proposed where weight quantization and full-precision weight retraining are performed iteratively, and the gap between the full-precision network and quantized network is supposed to diminish during the iterative process. Experiments are performed on both CNNs and LSTMs on some benchmark data sets.**

**Rating:** 5
**Confidence:** 4

**Review:**

The paper is a little hard to follow and some parts are poorly written. While the authors claim that they use the greedy approach (in sec 2) for quantization where both B and \alpha are learned in a greedy way, it is not clear why there is convergence difference between the two as claimed by the authors in section 3.1. Moreover, the authors claimed faster convergence of B than \alpha because fewer bit clips are observed from the left subplot of Figure 2. However, this conclusion is not quite convincing because 1) on the right subplot of Figure 2, it seems that \alpha also becomes more stable after 80k iterations; 2) the fewer bit clips may comes from using a stepwise learning rate decay scheme. Thus, the motivation for using another phase to train the \alpha is not strong.

The iterative quantization approach has limited novelty. It is similar to many quantization methods like BinaryConnect and Xnor-Net, except that the quantization step is not done immediately after the BP and model updates, but after some iterations of full-precision training. Moreover, these methods also use full-precision weights for update during training.

Clarity in the experiments section is a little better than the previous sections. However,
- The proposed method only performs comparably with TTQ, and shows significant accuracy drop on the Cifar-10 and Cifar-100 datasets (especially on Cifar-100)
- On the ImageNet dataset, there is a large accuracy drop of the proposed method compared to Zhou et al. (2018). Though the authors said that they believe their proposed model can reach a higher accuracy by using layer-by-layer quantization as in Zhou et al. (2018), it is hard to verify this claim due to lack of the corresponding results. Thus, efficacy of the proposed method on large datasets or models are hard to evaluate.

---

> ### Author Response · Authors · 2018-11-15
> **reply**
>
> Thank you for the review.
>
> B and \alpha both are learned greedily. B represents the sign of the weights and \alpha gives the value. As the training converges, the updates to the weights get smaller and smaller. This makes it more difficult for the weight to flip their sign bit, resulting in faster convergence of B.
> As \alpha belongs to floating point space, small updates are important for the convergence of \alpha. However, as B belongs to binary space, small updates do not affect B due its coarse-grained binary space, leading to faster convergence of B.
>
> Phase II (just train \alpha) reduces the computation overhead significantly. Phase II reduces the number of trainable parameters (approx. 1/1000 of the total parameters are trained) resulting in fewer parameter updates. Further, phase II does not need to perform the quantized-distortion step. Also, the forward-backward propagation are faster as the weights are represented in binary format (B, \alpha) instead of full precision weights.
>
> The proposed method differs from BinaryConnect/Xnor-net in the back-propagation step. Xnor-net evaluate the loss using binary weights and applies the updates to full-precision weights. Xnor-net performs quantization every step, which requires them to do binary backward propagation (needing a major change in training procedure).
>
> On cifar10 and cifar100 experiments with WideResNet, our method recovers full accuracy with its baseline (baseline is taken from Tensorflow models implementation given in https://github.com/tensorflow/models) as given in Table 2. McDonnell (2018) are unable to reach their baseline (although they have a higher baseline because they changed the model structure and training).
>
> On imagenet, our method gets comparable results with Xnor-net. Zhou et al. (2018) performs training for each layer separately requiring them to increase the number of forward propagation to be done. This scale the effective number of epochs by the number of layers. For resnet32, this scales the effective epochs by 32x. In contrast, the proposed method performs the quantization training in the original training time. As the training time is too large with layer-by-layer training for large networks, we have targeted feasibility of our method over improved results.

---

> > ### Comment · AnonReviewer1 · 2018-12-08
> > **post-rebuttal**
> >
> > I thank the authors for their response. However, the reply does not fully address my concerns. First, the authors claimed in their rebuttal that the convergence difference of B and \alpha is due to the different spaces they lie in (i.e., B in the binary space and \alpha in the continuous space). Empirically, B and \alpha indeed have similar convergence speed as shown in Figure 2. While I understand that training \alpha is efficient, but the authors do not directly answer why it is necessary to train \alpha additionally. I also do not see why the proposed method is computationally more advanced than BinaryConnect/Xnor-Net from the rebuttal, as one only needs to do a simple and efficient binarization before the forward propagation for BinaryConnect/Xnor-Net, while the proposed method requires several alternating training phases and is more complicated.
> >
> > Hence, I keep my rating unchanged.

---

> > > ### Author Response · Authors · 2018-12-10
> > > **Our response**
> > >
> > >
> > > Thank you for the post-rebuttal.
> > >
> > > Figure 2. (left) shows how often B values are flipped while Figure 2. (right) describes that 'alpha' still needs to be updated even after 80 iterations (when B values are not updated any more). Both alpha and 'B' values are updated during the exploration phase with high learning rate while 'alpha' updating without updating 'B' values is done with large learning rate at later phase as a fine-tuning procedure.
> > > Admittedly, even normal optimization process takes its most of time for fine-tuning within a local optimum space and it gives us the motivation of updating 'B' only for reduced quantization computation overhead when the training procedure gets into the fine-tuning process (without exploring multiple local minima).
> > > Please note that we observe similar frequencies of bit flipping in Figure 2 even without quantization or different quantization methods.
> > >
> > > Compared to BinaryConnect/Xnor-Net, we have the following advantages.
> > > 1) We do not perform the quantization process at every mini-batch, instead ours quantize the model occasionally according to the quantization step size.
> > > 2) After entering fine-tuning stage, we simplify the quantization model using 'alpha' only.
> > >
> > > On top of efficient computation method, our techniques achieve better accuracy (especially with RNN models and wide CNN models).

---

> > > ### Author Response · Authors · 2018-12-10
> > > **reply**
> > >
> > > Thank you for the post-rebuttal.
> > >
> > > Please note that the convergence of \alpha after 80k (be very small updates) is solely responsible for over 4% increase in quantized accuracy as can be seen in Figure 1.b.
> > >
> > > I would also like to address the simplicity of the algorithm:
> > > 1. Proposed quantization training method requires almost no change in the existing training procedure.
> > > 2. Phase1 performs quantization every few iterations. Phase2 directly trains \alpha using the quantized network.
> > > 3. As a result, the training time is also significantly reduced. Section 5.5 shows that our training method reduces quantization overhead from 40x to 10% of the training time in our experiments.

---

> > ### Public Comment · ~Mark_D_McDonnell1 · 2019-01-02
> > **"changed the model structure and training"**
> >
> > I may have misunderstood what exactly you meant when you wrote " McDonnell (2018) ... changed the model structure and training", but I want to clarify that there is *no difference* between architecture and training methods for baseline and 1-bit networks in McDonnell (2018). You appear to me to be saying this is not the case.  See point 2 on the strategy outlined on page 2: "Make minimal changes when training for 1-bit-per-weight" and Figure 2 in McDonnell (2018).

---

### Official Review · AnonReviewer2 · 2018-11-10
**Limited novelty and does not provide significant improvement compared to existing approaches**

**Rating:** 4
**Confidence:** 4

**Review:**

This work addresses the issue of quantization for neural network, and in particular focus on Ternary weight networks. The proposed approach has two phases, the first phase performs quantization and de-quantization at certain iterations during training, where the schedule of these operations are hyperparameters specified a priori. The second phase focuses on training the scaling factor. The first phase is similar to the iterative quantization method proposed in “Retraining-Based Iterative Weight Quantization for Deep Neural Networks”, and differs in that this work performs the quantization and de-quantization operations more frequently.
This work also proposed a modified version of L2 regularization, but it’s not clear how much benefit it provides compared to a regular L2 regularization. There is also a shuffling approach, but seems to provide limited improvement.
The experiment results in general does not provide convincing evidence that the proposed method outperforms existing approaches. For example, the ResNet-32 on CIFAR-10 result does not perform better than the one reported in “Trained ternary quantization”, and the ImageNet result is also worse than some existing works.
The work is lack of novelty and the results do not show significant improvement over existing approaches.

---

> ### Author Response · Authors · 2018-11-15
> **reply**
>
> Thank you for the review.
>
> In "Retraining-Based Iterative Weight Quantization for Deep Neural Networks", the concept of "exploration vs exploitation" was missing leading to longer training time (up to 6x of the original training time). Further, it also led to very low accuracy as we analysed the sensitivity of the quantization step size to the accuracy. We introduced a key hyper-parameter of the quantization step size, which determines the amount of weight distortion and the amount of retraining time (amount of convergence until the next step). Quantization step size was the key to make the technique work for CNNs.
>
> The L_KM loss function proposed helps reduce the variance of the weights compared to L2 loss, resulting in better approximation of \alpha. The biggest effect of L_KM loss is presented in "One-step Quantization" in section 5.3, where quantization is performed without any retraining. Using L_KM doubles accuracy compared to just using L2 loss (84.51% with L_KM compared to 44.33% with L2 loss) due to reduced variance of the weights sharing the same \alpha.
>
> There are many good quantization methods for CNN to use only 1bit or 2bit formats (including ternary). For CNN, our goal was to show our advantage in the computation complexity while training the quantized model (Our training method performs fewer quantization operations as analysed in section 5.5). However, for RNNs, it is still challenging to quantize with 1-2 bits (Table 3), where we perform better than the existing methods.
>
> Please refer to our response for AnonReviewer1 for our imagenet results in comparison with Zhou et al. (2018).

---

> > ### Comment · AnonReviewer2 · 2018-12-10
> > **thanks for the additional info**
> >
> > Thanks authors for providing more information. Even with these considered, the work still provided limited contribution, and therefore I would maintain the same rating.

---

### Meta-Review · Area_Chair1 · 2018-12-14
**Work would be strengthened by additional analyses**

**Confidence:** 4
**Recommendation:** Reject

**Metareview:**

The authors propose a technique for quantizing neural networks, which consist of repeated quantization/de-quantization operations during training, and the second step learns scale factors. The method is simple, clearly presented, and requires no change in the training procedure.
However, the authors noted that the work is somewhat incremental, and is similar to previously proposed approaches. As noted by the reviewers, the AC agrees that the work would be significantly strengthened by additional analysis of complexity in terms of computational time and memory relative to the other techniques.